# Receptor tyrosine kinases modulate distinct transcriptional programs by differential usage of intracellular pathways

Harish N Vasudevan, Pierre Mazot, Fenglei He, Philippe Soriano*

Department of Developmental and Regenerative Biology, Icahn School of Medicine at Mount Sinai, New York, United States

**Abstract** Receptor tyrosine kinases (RTKs) signal through shared intracellular pathways yet mediate distinct outcomes across many cell types. To investigate the mechanisms underlying RTK specificity in craniofacial development, we performed RNA-seq to delineate the transcriptional response to platelet-derived growth factor (PDGF) and fibroblast growth factor (FGF) signaling in mouse embryonic palatal mesenchyme cells. While the early gene expression profile induced by both growth factors is qualitatively similar, the late response is divergent. Comparing the effect of MEK (Mitogen/Extracellular signal-regulated kinase) and PI3K (phosphoinositide-3-kinase) inhibition, we find the FGF response is MEK dependent, while the PDGF response is PI3K dependent. Furthermore, FGF promotes proliferation but PDGF favors differentiation. Finally, we demonstrate overlapping domains of PDGF-PI3K signaling and osteoblast differentiation in the palate and increased osteogenesis in FGF mutants, indicating this differentiation circuit is conserved in vivo. Our results identify distinct responses to PDGF and FGF and provide insight into the mechanisms encoding RTK specificity.

*For correspondence: philippe.soriano@mssm.edu

## Introduction

Receptor tyrosine kinases (RTKs) signal through a shared set of intracellular pathways, including extracellular signal-related kinase (ERK) and phosphatidylinositol 3-kinase (PI3K), yet the in vivo functions directed by different RTKs can be quite distinct, raising the question of how specific cellular responses are elicited (*Lemmon and Schlessinger, 2010*). Several models have been put forth to explain how RTKs encode specificity (*Hunter, 2000*; *Simon, 2000*; *Pawson, 2004*; *Volinsky and Kholodenko, 2013*). In concept, distinct responses may be encoded by modulation of individual pathways downstream of receptor activation, with each pathway regulating a specific outcome. Alternatively, many cellular responses may require integration from multiple input pathways, and signal specificity could arise from this unique combination of pathways. Analysis of mice harboring point mutations to disrupt binding of specific effector proteins to RTKs suggests both these models may apply in vivo; platelet-derived growth factor (PDGF) Receptor α (*Pdgfra*) mutants display effector-specific phenotypes in line with the former model (*Klinghoffer et al., 2002*), but PDGF Receptor β (*Pdgfrb*) mediated outcomes require combined output across multiple pathways, consistent with the latter model (*Tallquist et al., 2003*). In addition, quantitative differences in the duration and magnitude of signal induction provide an added layer of regulatory complexity (*Marshall, 1995*). At the transcriptional level, one outcome of RTK activation is the expression of immediate early genes (IEGs) (*Cochran et al., 1983*; *Lau and Nathans, 1987*). Studies in cell culture have suggested IEGs constitute a generic readout of RTK activation with minimal specificity at the receptor or intracellular effector level (*Fambrough et al., 1999*), but genetic experiments in mice indicate a degree of IEG specificity (*Schmahl et al., 2007*). Therefore, a central goal remains to delineate RTK responsive

**eLife digest** Cells produce many different proteins that play a variety of important roles. For example, proteins called receptor tyrosine kinases can detect particular molecules and send signals to other parts of the cell to regulate the activity (or "expression") of genes involved in cell division, movement, and other processes.

Humans have 58 receptor tyrosine kinases, and defects in these proteins have been linked to diseases such as cancer and diabetes. However, many different receptors regulate the activities of shared sets of genes, so it is not clear how an individual receptor can specifically control the genes involved in a particular process.

Two receptor tyrosine kinases called PDGFR and FGFR are crucial for the development of the face, palate, and head in humans and other animals. Vasudevan et al. used a technique called RNA-sequencing to find out which genes are regulated by these receptors in mouse palate cells. The experiments show that there is a common set of genes whose activities change quickly—within 1 hour—in response to the activation of either PDGFR or FGFR. However, several hours later, cells in which PDGFR is activated have different patterns of gene expression compared to those with active FGFR.

Vasudevan et al. also found that FGFR promotes cell division, while PDGFR promotes the changing of palate cells into different types with more specialized roles. These different outcomes arise because PDGFR and FGFR use different signaling pathways that involve distinct proteins. For example, a protein called PI3K is critical for changes in gene expression in response to PDGFR but not FGFR.

These results suggest that PGDRF and FGFR control different cellular processes in the palate by sending distinct signals into the cell. Understanding the receptor tyrosine kinases and the networks of genes they activate will help us to identify the signals that are important for other processes, such as the development of the face.

transcriptional programs, identify the key signaling parameters encoding their regulation, and determine how these gene expression profiles dictate cellular decisions.

Many components of RTK signaling play important roles in mammalian craniofacial development (*Bentires-Alj et al., 2006*; *Newbern et al., 2008*; *Fantauzzo and Soriano, 2015*). In particular, PDGF and FGF signaling are both essential for midface development. In mice, loss of *Pdgfra* (*Soriano, 1997*) or its ligands *Pdgfa* and *Pdgfc* (*Ding et al., 2004*) results in facial clefting, and mice harboring a mutation abrogating PI3K binding to PDGFRα mirror these craniofacial phenotypes, implicating PI3K as the main effector of PDGFRα signaling (*Klinghoffer et al., 2002*). In addition, both *Pdgfra* (*Wnt1-Cre; Pdgfra^{fl/fl}*) (*Tallquist and Soriano, 2003*) and FGF receptor 1 (*Wnt1-Cre; Fgfr1*) (*Trokovic et al., 2003*; *Wang et al., 2013*) neural crest conditional mutants exhibit cleft face, indicating both pathways are required for normal development of the neural crest derived facial skeleton. At the intracellular pathway level, previous work has implicated ERK as a key effector downstream of FGF signaling (*Lanner and Rossant, 2010*). Furthermore, mutations in both PDGF and FGF signaling have been linked to craniofacial syndromes in humans (*Choi et al., 2009*; *Miraoui and Marie, 2010*; *Rattanasopha et al., 2012*). Interestingly, chimeric receptor experiments in mice have shown that the intracellular domain of *Fgfr1* cannot compensate for *Pdgfra* during development, suggesting these two receptors transmit biologically distinct signals in vivo (*Hamilton et al., 2003*). The midface thus offers a unique opportunity to interrogate the mechanisms of signal specificity between these two RTKs in a developmentally relevant system.

Given the requirement for PDGF and FGF signaling in the development of the neural crest derived midface, we sought to compare the gene expression programs regulated by these two RTKs. The architecture of the transcriptional response to RTK activation consists of three stereotypic waves: an IEG response involving core transcriptional regulators (*Fos*, *Jun*, *Egr*), a delayed response playing a feedback role (phosphatases, RNA-binding proteins), and a late sustained response determining cellular outcome (*Amit et al., 2007*; *Avraham and Yarden, 2011*). However, the degree of conservation between genes regulated in each wave across different RTK families is unclear. Furthermore, although classic feedback regulators of mitogen-activated protein kinase (MAPK)

pathways have been described, such as dual-specificity phosphatases (DUSPs) for ERK and c-Jun N-terminal kinase (JNK) (*Li et al., 2007*; *Owens and Keyse, 2007*), the extent of effector-dependent transcription genome-wide is not well characterized.

In the present work, we compare the transcriptional response to PDGF and FGF signaling in E13.5 mouse embryonic palatal mesenchyme (MEPM) cells. Although both PDGF and FGF are required in the neural crest for craniofacial development, we find distinct transcriptional programs, effector dependencies, and cellular outcomes in response to each RTK. While many genes in the early wave are shared across the two RTKs, FGF induces a quantitatively stronger response than PDGF. In addition, the feedback control provided by the delayed transcriptional wave displays distinct characteristics in response to PDGF and FGF. By exploring the effect of MEK/ERK and PI3K inhibition on these RTK-regulated gene expression profiles, we find PDGF-mediated transcription displays greater PI3K dependence, while FGF-mediated gene expression programs predominantly require ERK activity. This relationship is conserved at the level of cellular outcome, with FGF driving proliferation but PDGF promoting PI3K-dependent differentiation. Finally, we show overlapping domains of PI3K signaling, PDGF target gene expression, and skeletal differentiation during palatogenesis in vivo, a process perturbed in *Fgfr1* conditional mutants. Taken together, our studies suggest unique roles for PDGF and FGF during development of the facial skeleton, and more broadly, demonstrate that distinct transcriptional responses to RTK signaling are encoded through qualitative and quantitative differences in intracellular pathway activation.

## Results

### PDGF and FGF have distinct patterns of effector activation and transcriptional responses in E13.5 MEPMs

Since neural crest conditional loss of either *Fgfr1* or *Pdgfra* leads to clefting, we chose to perform RNA-seq on E13.5 MEPMs treated with either PDGFA or FGF1 + heparin to identify the gene expression programs regulated by each signaling pathway (*Figure 1A*). MEPMs express many essential markers of the palatal mesenchyme and have been previously used to study responses to many pathways (*Bush and Soriano, 2010*; *Iwata et al., 2012*; *Fantauzzo and Soriano, 2014*), including PDGF and FGF (*Vasudevan and Soriano, 2014*). We performed RNA-seq at 1 and 4 hr following ligand treatment in order to characterize both the early and late responses to PDGF and FGF signaling (*Supplementary File 1*). In the samples submitted for sequencing, both PDGF and FGF induced a robust phospho-ERK (pERK) response at 15 min (*Figure 1—figure supplement 1A*), and MEPMs generated from *Pdgfra-GFP* (*Hamilton et al., 2003*) and *Fgfr1-CFP* (to be described elsewhere) knockin reporter embryos display expression of each receptor at the protein level in all cells (*Figure 1—figure supplement 1B*), further validating MEPMs as a suitable system to study RTK responses.

We first plotted the expression of all genes with FPKM (fragments per kilobase of exon per million reads mapped) values >1 at both 1 hr (*Figures 1B* and 4 hr (*Figure 1B'*); although only a small number of genes are differentially regulated between the 1-hr PDGF and 1-hr FGF samples (Cuffdiff q < 0.1, *Supplementary File 2*; *Trapnell et al., 2010*), the difference in the response to these two growth factors is much greater at 4 hr. Consistent with this observation, visualization of all replicates by principal component analysis (PCA) (*Figure 1—figure supplement 1C*) revealed that the 1-hr PDGF and 1-hr FGF samples cluster together, but the 4-hr FGF replicates are distinctly separate from the 4-hr PDGF samples. Comparing the stimulated MEPMs to untreated cells, genes differentially regulated at 1 hr by either PDGF or FGF (*Supplementary File 2*) show high correlation ($r^2 = 0.8173$, *Figure 1C*), but by 4 hr, the two RTK signals are divergent ($r^2 = 0.2881$, *Figure 1C'*). In addition, the genes regulated by PDGF at 1 hr (n = 40) form a subset of those genes regulated by FGF at 1 hr (n = 159), further highlighting the similarity within the early response to both growth factors. Gene ontology analysis (*Huang et al., 2009*) of the genes induced at 1 hr revealed an enrichment of transcription factors and MAP kinase phosphatases downstream of both RTKs (*Figure 1—figure supplement 1D*, p < 0.001), similar to previous descriptions of the response to RTK activation (*Amit et al., 2007*; *Avraham and Yarden, 2011*). To better visualize the organization of these targets, we constructed a protein–protein interaction (PPI) network from the genes regulated at 1 hr; in constructing this network, we only included direct interactions between proteins (path length = 1) and excluded predicted interactions (*Berger et al., 2007*; *Chen et al., 2012*). The resulting network

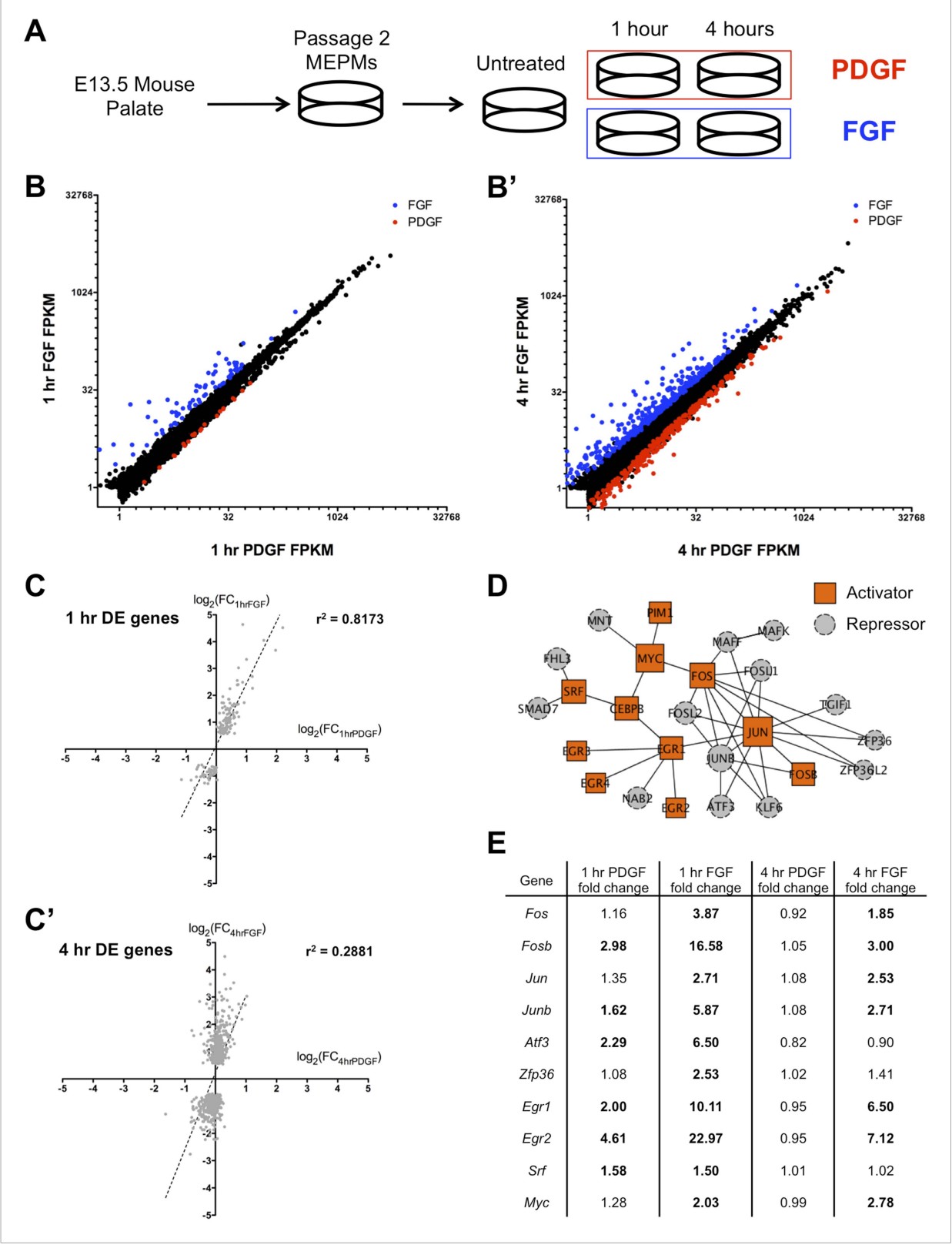

**Figure 1**. FGF and PDGF stimulation result in distinct transcriptional responses. (**A**) Mouse embryonic palatal mesenchyme (MEPM) cells were dissected from E13.5 embryos, passaged twice, and then serum starved overnight prior to stimulation with either PDGFAA or FGF1 + heparin. (**B**) Expression of all genes with FPKM (fragments per kilobase of exon per million reads mapped) >1 at 1 hr (11,217 genes) and (**B'**) 4 hr (11,266 genes). Genes colored blue are

*Figure 1. continued on next page*

*Figure 1. Continued*

significantly increased with fibroblast growth factor (FGF) treatment and genes colored red are significantly increased with platelet-derived growth factor (PDGF) treatment. Values plotted on log$_2$ scale. (**C**) Fold change (FC) comparison for all differentially expressed (DE) genes at (**C**) 1 hr or (**C'**) 4 hr (compared to serum starved sample) shows high correlation between the transcriptional response to each growth factor at 1 hr but low correlation at 4 hr. (**D**) Protein–protein interaction (PPI) network for all genes upregulated at 1 hr contains many classic immediate early genes (such as AP-1 components, *Myc*, and *Srf*). Genes are colored based on their primary reported role in transcriptional regulation (22, 23), with orange squares representing activators and gray circles representing repressors. (**E**) FC (compared to untreated sample) for selected genes upregulated at 1 hr. Genes in bold are induced >1.5-fold in response to the indicated growth factor. Although both PDGF and FGF regulate many shared targets, the induction in response to FGF exhibits greater magnitude (*Fos*, *Fosb*, *Junb*, *Atf3*, *Egr1*, *Egr2*) and longer duration (*Fos*, *Fosb*, *Jun*, *Junb*, *Egr1*, *Egr2*).

The following figure supplement is available for figure 1:

**Figure supplement 1**. Characterization of E13.5 MEPMs and transcriptional response to PDGF and FGF signaling.

contains 25 upregulated genes (out of 113 total induced genes), including many classic components of the IEG response, such as activator protein-1 (AP-1) subunits (*Fos*, *Jun*) and their regulators (*Zfp36*, *Atf3*) as well as *Myc*, *Egr1-4*, and *Srf* (***Figure 1D***). Closer inspection revealed that many shared target genes within this network are regulated to both a stronger magnitude and longer duration following FGF treatment compared to PDGF treatment (***Figure 1E***). Indeed, *Fos*, *Fosb*, *Jun*, and *Junb* all exhibit differences in signal magnitude and/or duration in response to FGF, as validated by qPCR (***Figure 1—figure supplement 1E***). In sum, the two RTKs induce a similar gene expression profile at 1 hr (***Figure 1C***), as evidenced by the high-correlation coefficient and overlap between genes differentially regulated by PDGF and FGF at 1 hr. However, FGF drives a quantitatively stronger early response with many transcription factors showing both a stronger magnitude and greater duration of induction in response to FGF compared to PDGF (***Figure 1E***), which may explain in part the divergent gene expression profiles observed at 4 hr.

## Differential roles for the delayed transcriptional response in regulation of signaling downstream of FGF and PDGF

Given the importance of MEK/ERK and PI3K/Akt signaling downstream of these RTKs during development (***Klinghoffer et al., 2002***; ***Corson et al., 2003***; ***Lanner and Rossant, 2010***; ***Fantauzzo and Soriano, 2014***), we analyzed pERK and pAkt activation following FGF and PDGF stimulation in MEPMs (***Figure 2A***). Consistent with the stronger response to FGF treatment in the gene expression data, the FGF-induced pERK response displays both a higher magnitude and longer duration of activation compared to the PDGF-induced pERK signal. In contrast, both FGF and PDGF induce similar patterns of pAkt activation, but the magnitude of the PDGF-pAkt induction is slightly greater. The FGF-pERK signal is apparent up to 6 hr following growth factor treatment (***Figure 2—figure supplement 1A***), and increasing the dose of PDGFA ligand did not alter the kinetics of pERK activation (***Figure 2—figure supplement 1B***). To better understand differences in the response to PDGF and FGF, we performed gene ontology analysis for the molecular function of genes that are differentially expressed (DE) between 4-hr PDGF treatment and 4-hr FGF treatment (***Figure 2B***, p < 0.001) (***Huang et al., 2009***). The top results for genes enriched following FGF treatment are sets associated with modulation of signaling, such as protein kinases and GTPase regulators, which may function as activators of Ras to promote MEK/ERK signaling. Transcriptional feedback regulation of RTK signaling is well established, particularly the role of DUSPs providing negative feedback for MAPK signaling (***Amit et al., 2007***; ***Li et al., 2007***; ***Owens and Keyse, 2007***). Indeed, many DUSPs (MAPK phosphatases) are induced in response to both PDGF and FGF treatment at 1 hr (***Figure 1—figure supplement 1D***), but FGF alone induces the expression of kinases and GTPase regulators at 4 hr, suggesting a distinct role for the FGF response in regulating MEK/ERK activity. We, thus, performed Western blots in the presence of cycloheximide following both PDGF and FGF treatment to determine the effect of inhibiting protein synthesis (and consequently, the delayed transcriptional response) on ERK activation. Consistent with previous work exploring the role of DUSP-mediated negative feedback (***Amit et al., 2007***), cycloheximide treatment increased the duration of the PDGF-pERK response (***Figure 2C***). However, we observed the opposite effect of cycloheximide treatment on the FGF-pERK response (***Figure 2C'***), suggesting a positive feedforward loop in which

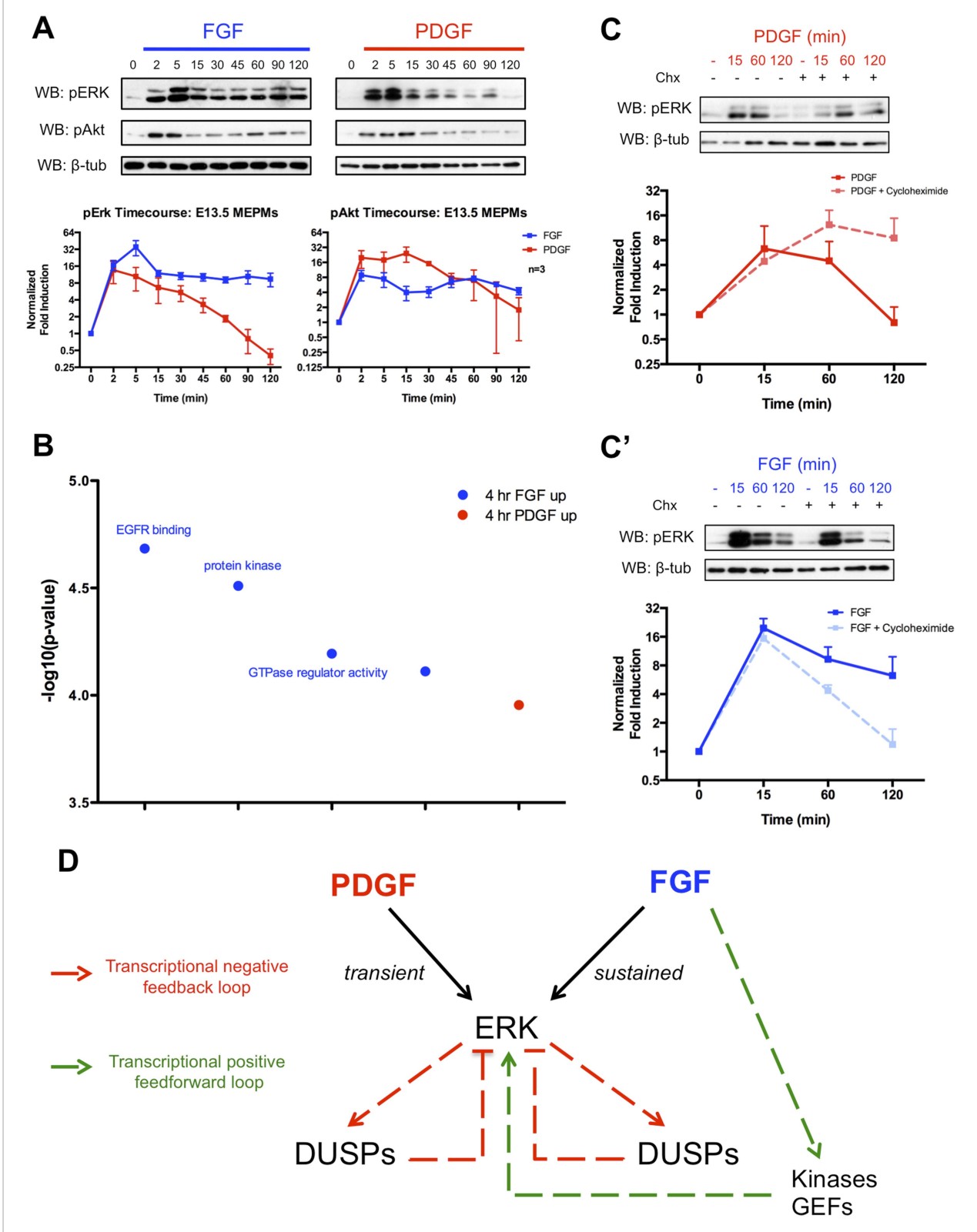

**Figure 2**. The delayed transcriptional response provides differential regulation of pERK duration in response to FGF and PDGF signaling. (**A**) Signaling time course shows a more robust phospho-ERK (pERK) response to FGF (blue) than PDGF (red) and a similar pAkt response to both growth factors. (**B**) Gene ontology analysis (molecular function) of genes DE between the 4-hr PDGF and 4-hr FGF conditions indicates enrichment for kinases and GTPase

*Figure 2. continued on next page*

*Figure 2. Continued*

regulators in response to FGF signaling. (**C**) Cycloheximide treatment has opposite effects on pERK duration following (**C**) PDGF and (**C′**) FGF stimulation, indicating the delayed transcriptional response (dependent on protein synthesis and thus inhibited by cycloheximide) can provide both negative and positive signals to modulate pERK kinetics. (**D**) Model depicting loops that regulate the duration of the pERK wave in response to receptor tyrosine kinase (RTK) signaling includes both negative (dual-specificity phosphatases [DUSPs]) and positive (kinases, GEFs) components from the delayed transcriptional response. Western blot quantification plotted as mean ± SEM, n = 3.

The following figure supplement is available for figure 2:

**Figure supplement 1**. Signaling kinetics and organization of the late transcriptional response in response to PDGF and FGF treatment.

FGF induces the expression of kinases and GEFs to modulate the ERK response in addition to activating ERK directly. To further explore the architecture of the late FGF response, we constructed a PPI network from genes increased at 4-hr FGF stimulation compared to 4-hr PDGF treatment. The FGF network (*Figure 2—figure supplement 1C*) contains *Prkca* as a highly connected node, which is interesting given reported roles for protein kinase C (PKC) in facilitating a sustained pERK response (*Bhalla et al., 2002*; *Santos et al., 2007*) as well as its importance downstream of FGF in skeletal development (*Miraoui and Marie, 2010*). We found that inhibition of PKC decreased both the initial pulse and sustained activation of the FGF-mediated pERK response in MEPMs (*Figure 2—figure supplement 1C′*), consistent with its potential function as a hub within the FGF response network. In addition, many of the kinases and GEFs transcriptionally regulated by FGF have reported roles in positively modifying MEK/ERK signaling (*Figure 2—figure supplement 1D*), which may explain in part the residual pERK response to FGF in the presence of PKC inhibition. Collectively, these data support a model in which the balance between positive and negative transcriptional loops is critical for determining patterns of pERK response to different RTKs (*Figure 2D*).

## The FGF transcriptional response is primarily MEK/ERK dependent, while the PDGF response shows greater PI3K usage

The differences in signaling pathway activation following PDGF and FGF stimulation led us to consider how inhibition of these pathways affected the two RTK-mediated transcriptional programs. Thus, we analyzed the effector dependence of the transcriptional response by performing RNA-seq in cells stimulated with either PDGF or FGF in the presence of PD325901 (MEK inhibitor) or LY294002 (PI3K inhibitor) (*Supplementary File 1*). PCA on all thirteen sequenced conditions (*Figure 3—figure supplement 1A*) segregates the samples based on growth factor treatment along PC1 (44.57% of the variance) and on inhibitor treatment along PC2 (17.72% of the variance). Similarly, the correlation matrix for all sampled replicates mimics the PCA, with the 4 hr FGF and 4 hr FGF + LY showing a gene expression profile distinct from all other samples (*Figure 3—figure supplement 1B*). We next directly tested the effect of pathway inhibition on both the shared gene expression program induced by the two RTKs at 1 hr (113 genes total) and the genes regulated between FGF and PDGF treatment at 4 hr. Globally, FGF target genes show greater MEK/ERK dependence than PI3K dependence, while PDGF responsive genes exhibit the opposite relationship (*Supplementary File 3*) mirroring the reported signaling requirements for each RTK. This trend is apparent at 1 hr (*Figure 3A, A′*) and striking for genes DE at 4 hr, where 52% of FGF responsive genes are MEK/ERK dependent (*Figure 3B*) but only 22% were PI3K dependent (*Figure 3B′*). This dependence is inverted for PDGF, where 9% are MEK/ERK dependent (*Figure 3C*) but 28% are PI3K dependent (*Figure 3C′*). Strikingly, MEK inhibition increases the expression of PDGF targets (*Figure 3C*) but not FGF targets (*Figure 3B*), while PI3K inhibition increases the expression of FGF responsive genes (*Figure 3A′, B′*), suggesting the FGF-ERK and PDGF-PI3K relationships are important for both gene induction and repression. These intracellular pathway dependencies are independent of the magnitude of induction/repression, as varying the FC threshold did not affect the PDGF-PI3K or FGF-ERK relationships (*Figure 3—figure supplement 1C, D′*).

We then considered the degree of overlap between ERK and PI3K targets. At 1 hr, many shared target genes show similar ERK or PI3K dependence, possibly constituting a core pathway specific gene set (*Figure 3D*). On the other hand, cross-comparison of PI3K- and ERK-dependent genes reveals some overlap (*Figure 3D′*), underscoring a degree of plasticity in pathway usage downstream

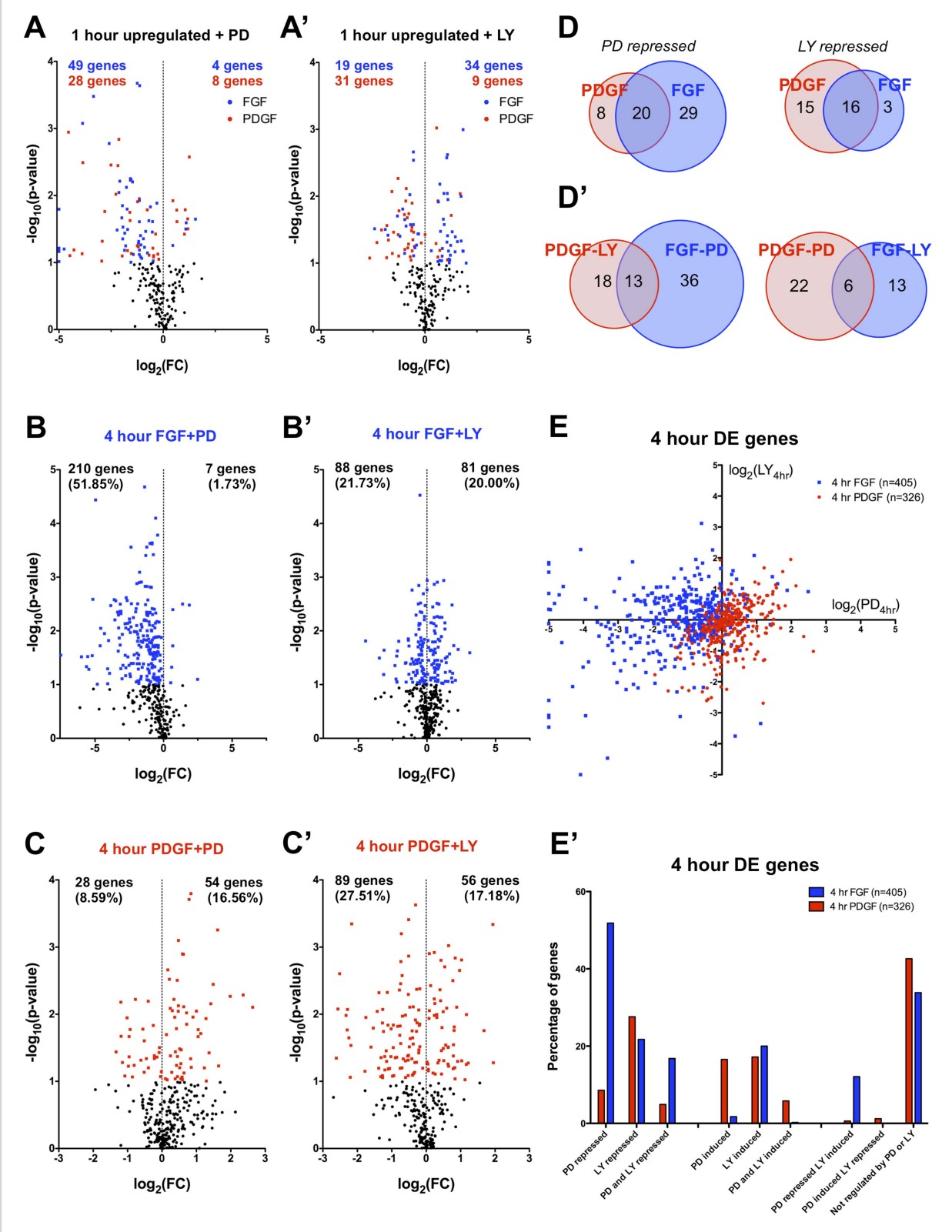

**Figure 3**. FGF and PDGF transcriptional responses exhibit differential usage of intracellular pathways. (**A**) Volcano plots visualizing the effect of (**A**) MEK (Mitogen/Extracellular signal-regulated kinase) inhibition and (**A'**) phosphatidylinositol 3-kinase (PI3K) inhibition on the shared set of 113 genes upregulated at 1 hr reveal that FGF (blue points) shows increased dependence on MEK/extracellular signal-related kinase (ERK) signaling, while PDGF

*Figure 3. continued on next page*

*Figure 3. Continued*

(red points) utilizes PI3K to a greater degree. X-axis plotted as $\log_2$([1 hr ligand + inhibitor]/[1 hr ligand]). (**B**) Genes with increased expression at 4-hr FGF treatment show higher dependence on (**B**) MEK/ERK activity compared to (**B'**) PI3K. (**C**) In contrast, genes with increased expression at 4-hr PDGF treatment show greater usage of (**C'**) PI3K compared to (**C**) MEK/ERK. X-axis plotted as $\log_2$([4 hr ligand + inhibitor]/[4 hr ligand]). Data analyzed using two sample t-test, and genes at $p < 0.1$ are colored significant. Black points represent genes not significant at this threshold in all plots. (**D**) A core set of MEK/ERK (20 genes) and PI3K (16 genes) are dependent on these pathways downstream of both PDGF and FGF. (**D'**) A minority of genes can be activated through either MEK/ERK or PI3K signaling in response to PDGF or FGF, indicating a degree of plasticity in intracellular pathway usage. (**E**) Scatter plot comparing effect of MEK/ERK and PI3K inhibition on all 4 hr DE genes reflects FGF-ERK and PDGF-PI3K dependencies. Data plotted as $\log_2$([4 hr ligand + inhibitor]/[4 hr ligand]) and capped at ±5 for visualization. (**E'**) 52% of FGF target genes at 4 hr are repressed by MEK/ERK inhibition, while 28% of PDGF responsive genes are repressed by PI3K inhibition. Interestingly, 12% of genes are repressed by MEK/ERK inhibition and 'superinduced' by PI3K inhibition, indicating crosstalk between these pathways. Furthermore, 43% of PDGF responsive genes and 34% of FGF responsive genes are not significantly affected by either inhibitor, which suggests either combinatorial requirement of MEK and PI3K or alternate intracellular pathways drive expression of these genes.

The following figure supplement is available for figure 3:

**Figure supplement 1**. RTK target genes show distinct patterns of effector dependence in a threshold-independent manner.

of different RTKs. When directly comparing the effect of MEK and PI3K inhibition at 4 hr (*Figure 3E*), the crosstalk between FGF targets is particularly striking, as a large number of genes repressed by MEK inhibition were regulated (both positively and negatively) by PI3K inhibition. Furthermore, while 23% of PDGF targets are specifically PI3K dependent, only 5% of FGF targets are repressed by PI3K alone (*Figure 3E'*). Instead, many FGF-PI3K targets are also repressed by MEK inhibition (77%), and conversely, a number of FGF-ERK targets are induced by PI3K inhibition (23%), indicating the effect of PI3K inhibition downstream of FGF involves crosstalk with ERK signaling.

## Inhibition of intracellular effector activation leads to induction of alternate signaling pathways

We were intrigued by the subset of genes exhibiting significantly increased expression ('superinduction') following inhibitor treatment at both 1 and 4 hr. Both RTKs show increased *Jun* expression in response to PD325901, while FGF and LY294002 treatment upregulates many classic ERK targets such as *Fos*, *Fosb*, and *Dusp6* (*Supplementary File 3*), suggesting this 'superinduction' may reflect compensatory activation of other intracellular pathways. Many such examples of crosstalk between ERK and PI3K have been documented (*Mendoza et al., 2011*). Thus, we assayed the activation of these pathways following MEK inhibition with PD325901 and PI3K inhibition with LY294002. We found striking induction of pJNK upon pERK inhibition by PD325901 downstream of both FGF (*Figure 4A*) and PDGF (*Figure 4B*) stimulation. Interestingly, a moderate change in pAkt induction following PDGF treatment and MEK inhibition seems apparent, although this induction was not observed at the dose used for the RNA-seq experiment (1 µM PD325901). Next, we investigated the effect of PI3K inhibition with LY294002. We found that FGF-mediated pERK induction increased with LY294002 treatment (*Figure 4A'*), but PDGF treatment did not produce this increase in pERK signal (*Figure 4B'*), consistent with changes observed at the level of target gene expression. At the inhibitor doses used in the RNA-seq experiment (1 µM PD325901, 10 µM LY294002), both PDGF and FGF significantly induce JNK activation in the presence of MEK inhibition, while only FGF activates ERK when PI3K is inhibited (*Figure 4C, C'*). Finally, we performed qPCR for a subset of target genes to confirm their response to pathway inhibition. *Fos*, *Fosb*, and *Junb* exhibit 'superinduction' downstream of FGF specifically in response to PI3K inhibition, confirming these genes are indeed MEK/ERK dependent and providing further evidence that LY294002-mediated induction of pERK can drive transcriptional changes (*Figure 4D*). Similarly, *Jun* is 'superinduced' in the presence of PD325901 downstream of both RTKs. There is a modest increase in *Jun* induction following FGF treatment with PI3K inhibition, likely reflecting the compensatory induction of pERK in this condition and consequent crosstalk between ERK and JNK signaling. In addition, *Fos* induction is increased with LY294002 treatment even in the absence of growth factor, which may reflect a degree of growth factor independent crosstalk between PI3K and ERK. However, this effect synergizes with ligand treatment, indicating this compensation across intracellular pathways is at least partially dependent on receptor activation.

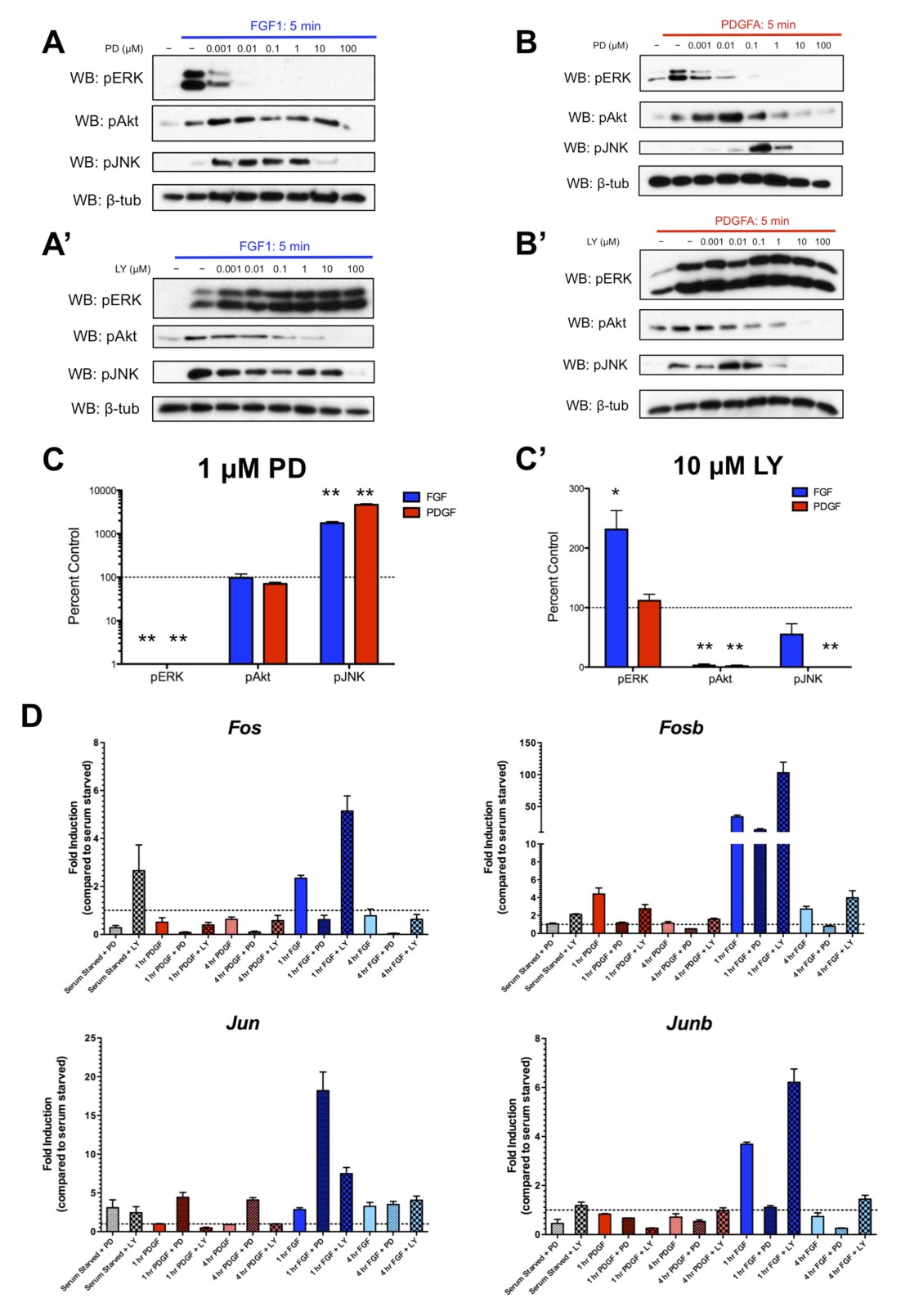

**Figure 4.** Inhibition of effector activation results in compensatory induction of alternate signaling pathways detectable in the transcriptional response. (**A**) PD325901 dose response Western blots reveal induction of pJNK as pERK is progressively inhibited downstream of FGF signaling. (**A'**) Similarly, LY294002 dose response Western blots reveal increased pERK signal as pAkt is inhibited. (**B**) Inhibitor dose response Western blots in response to PDGF signaling show activation of c-Jun N-terminal kinase (JNK) in response to MEK/ERK inhibition but (**B'**) no activation of ERK following PI3K inhibition. *Figure 4. continued on next page*

Figure 4. Continued

(**C**) Quantification of effector activation in response to FGF (blue) or PDGF (red) at the doses used in the RNA-seq experiment reflects (**C**) increased pJNK activation when MEK/ERK signaling is inhibited and (**C'**) increased pERK induction when PI3K activity is blocked. Data plotted as mean ± SEM, n = 3 and compared using two sample, unpaired t-test to baseline of 100% (no change). *p < 0.05; **p < 0.001. (**D**) Gene expression reflects the crosstalk observed at the signaling level, as verified by qPCR for selected target genes. Canonical ERK targets such as *Fos*, *Fosb*, and *Junb* are 'superinduced' upon LY treatment, while the JNK target *Jun* is 'superinduced' with PD treatment. Interestingly, a degree of 'superinduction' is observed in the presence of inhibitor prior to growth factor addition, which may reflect RTK-independent crosstalk between intracellular pathways. Data plotted as mean ± SEM, n = 3.

## FGF drives MEK/ERK-dependent cell proliferation, and PDGF promotes PI3K-dependent cell differentiation

We next investigated the functional consequence of these differential effector activation patterns and transcriptional programs in response to PDGF and FGF signaling. Gene ontology for biological processes (*Huang et al., 2009*) enriched in genes differentially regulated by PDGF and FGF at 4 hr revealed an FGF-mediated proliferation program and a PDGF-regulated differentiation circuit (*Figure 5A*, p < 0.001). Interestingly, we also observed enrichment for regulators of Wnt signaling in the 4-hr PDGF condition, consistent with reports of antagonism between Wnt and FGF as regulating the balance between differentiation and proliferation in skeletal development (*Mansukhani et al., 2005*). We further identified a set of differentiation genes (*Id1*, *Id2*, *Id3*, *Mef2c*, *Atoh8*) in the PPI network constructed from genes regulated by PDGF treatment at 4 hr (*Figure 5—figure supplement 1A*) and confirmed these genes by qPCR (*Figure 5—figure supplement 1A'*). In line with PDGF directing a skeletal differentiation program, the *Id* genes (*Maeda et al., 2004*; *Kee and Bronner-Fraser, 2005*) and *Mef2c* (*Verzi et al., 2007*) are required for craniofacial skeleton development in vivo. In addition, mouse genome informatics mammalian phenotype analysis to determine overrepresented mouse phenotypes (*Chen et al., 2013*) identifies abnormal craniofacial bone development as the most enriched phenotype in the 4-hr PDGF target genes (*Figure 5—figure supplement 1B, B'*). Plotting the expression of these FGF-proliferation and PDGF-differentiation genes emphasizes the distinct responses of these gene sets to these two growth factors (*Figure 5B*). Furthermore, many proliferation genes are reduced upon MEK inhibition and induced upon PI3K inhibition, while the opposite is apparent for differentiation genes, consistent with the FGF-ERK and PDGF-PI3K dependencies observed globally at the transcriptional level.

To directly assay cell proliferation, we performed BrdU labeling at 4 hr following either FGF or PDGF treatment in E13.5 MEPMs; while PDGF induces a modest response compared to serum-starved cells, we found a significantly greater proliferative response to FGF (*Figure 5C*). Crystal violet staining for cell viability confirms a greater effect of FGF than PDGF as well as the importance of MEK/ERK activity for this response (*Figure 5—figure supplement 1C*). Furthermore, the PDGF-dependent effect on cell viability is significantly greater at day 3 compared to the 0.1% fetal bovine serum (FBS) treated cells, underscoring the role of PDGF in cell survival/growth in the absence of other growth factors. We next measured apoptosis following PDGF and FGF treatment in the presence of both inhibitors (*Figure 5—figure supplement 1D*). Consistent with the cell viability results and phospho-JNK induction patterns, MEK inhibition following FGF stimulation results in a greater increase in apoptosis than inhibition of FGF-mediated PI3K signaling; in contrast, PI3K inhibition has a greater effect than MEK inhibition downstream of PDGF stimulation, although inhibiting either pathway results in increased apoptosis. Finally, we tested cell differentiation by alkaline phosphatase (AP) staining, a marker of osteoblast differentiation (*Wu et al., 2008*). PDGF-treated MEPMs display a robust AP response, while FGF-treated cells show a striking lack of AP positive cells (*Figure 5D*). In addition, MEK/ERK inhibition increases AP staining, but PI3K inhibition represses osteoblast differentiation (*Figure 5D*). These experiments suggest the following model: FGF drives cell proliferation and represses cell differentiation in a MEK/ERK-dependent manner, while PDGF facilitates cell differentiation, at the expense of reduced proliferation, in a PI3K-dependent manner.

## The FGF repressed, PI3K-dependent differentiation circuit is conserved during mouse craniofacial development

Finally, we sought to investigate the FGF-ERK-proliferation and PDGF-PI3K-differentiation axes in vivo. We first examined the expression pattern of *Fgfr1* in relation to *Dusp6* (ERK-specific), *Dusp1*

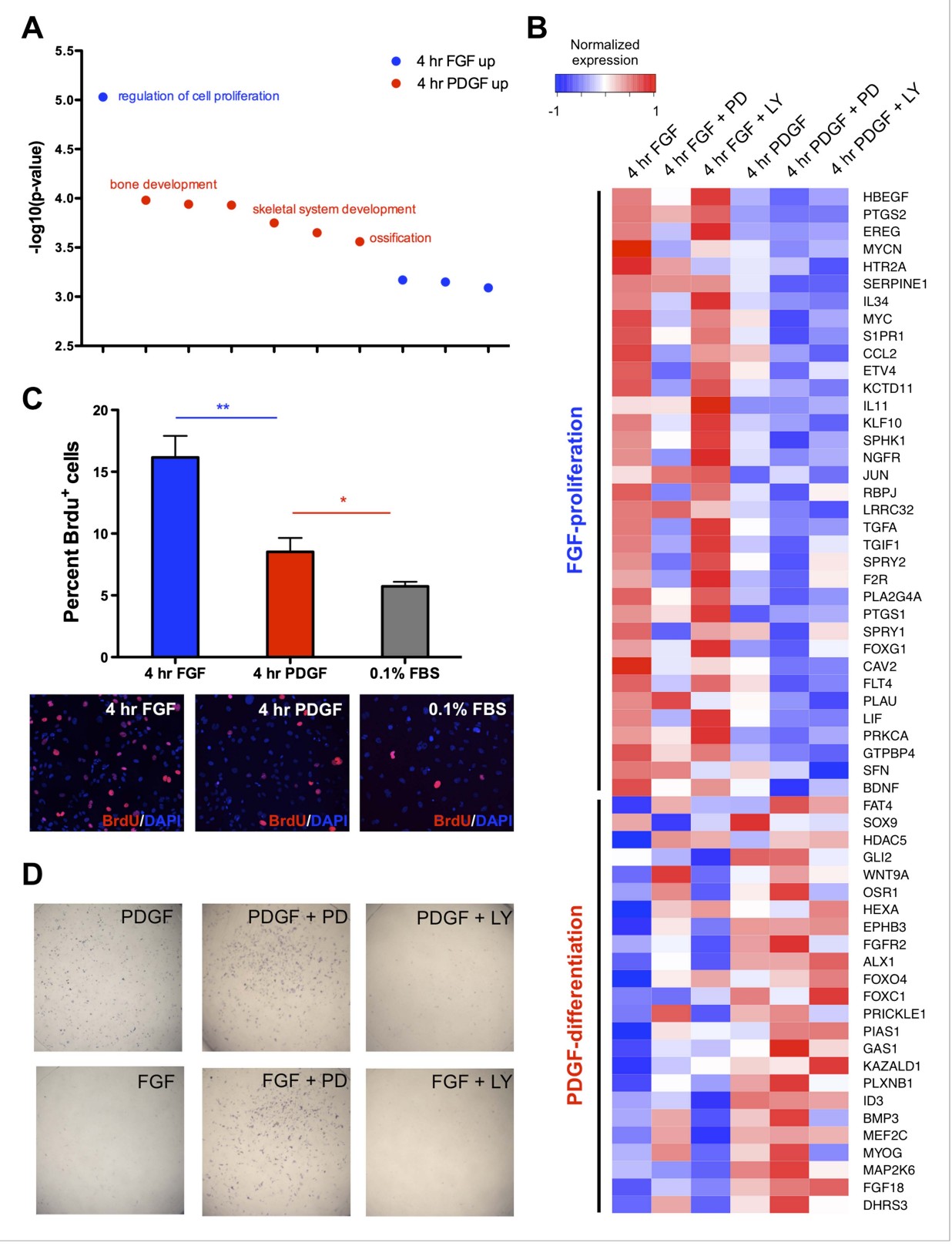

**Figure 5**. Distinct cellular outcomes are specified in response to PDGF and FGF signaling. (**A**) Gene ontology analysis of genes DE between the 4-hr FGF and 4-hr PDGF conditions shows enrichment for regulators of cell proliferation in genes upregulated by FGF treatment. In contrast, genes implicated in skeletal differentiation are overrepresented in the genes more highly expressed following PDGF treatment (**B**) Genes associated with cell proliferation are

*Figure 5. continued on next page*

*Figure 5. Continued*

strongly upregulated by FGF treatment, while genes associated with cell differentiation are increased following PDGF stimulation. In addition, MEK inhibition represses many proliferation genes but induces differentiation genes, while PI3K inhibition has the opposite effect. Genes ordered by decreasing ratio of FGF:PDGF expression. (C) FGF induces a significantly more robust proliferation response than PDGF in MEPMs, although PDGF does promote a modest response compared to starved cells (0.1% FBS). Quantification plotted as mean ± SEM, n = 3. Two tailed, unpaired t-test: *$p < 0.05$; **$p < 0.005$ (D) PDGF, but not FGF, treatment promotes alkaline phosphatase (AP) (osteoblast marker) positive cells. Furthermore, PD treatment drives AP staining, while LY treatment abolishes AP staining independent of growth factor stimulation. AP staining performed 8 hr following ligand treatment.

The following figure supplement is available for figure 5:

**Figure supplement 1**. PDGF-mediated differentiation responses exhibit a preference for PI3K signaling, while FGF-mediated effects on proliferation show greater MEK/ERK dependence.

(JNK-specific), and *Dusp10* (JNK-specific) in the E13.5 palate by in situ hybridization (*Figure 6—figure supplement 1A*). Although *Dusp1* and *Dusp10* are primarily epithelial, *Fgfr1* and *Dusp6* are co-expressed in the anterior palatal mesenchyme, consistent with previous work implicating ERK in proliferation within this region (*Bush and Soriano, 2010*). Similarly, immunohistochemistry (IHC) revealed pERK is scattered in the anterior palatal mesenchyme, with some epithelial staining (*Figure 6—figure supplement 1B*). However, cell proliferation along the anterior–posterior axis is relatively uniform in the E13.5 palate (*Bush and Jiang, 2012*), complicating assignment of the observed expression patterns to a spatially restricted proliferation program. We next explored the relationship between PI3K signaling and osteoblast differentiation. Whole mount IHC for pAkt at E13.5 demonstrates expression restricted to the developing upper lip and middle to posterior palate, overlapping with AP in these structures (*Figure 6A*). We further assayed the pattern of Runx2 expression (*Figure 6B*) in comparison to *Pdgfra*, *Id1*, and *Id3* in the E13.5 palate (*Figure 6B'*), finding shared domains of expression and exclusion in the palate. Taken together, this correlation between PI3K activity, the *Id* genes, and regions of osteoblast differentiation supports the existence of a spatially restricted PDGF-PI3K-differentiation axis.

In MEPMs, FGF drives proliferation and represses differentiation. One prediction from this observation is that *Fgfr1* mutants would have decreased repression of this program, and consequently, increased differentiation in the midface. We, therefore, performed AP staining on neural crest conditional *Fgfr1* mutants (*Wnt1-Cre; Fgfr1^{fl/fl}*) at E14.5 to investigate defects in osteoblast differentiation. We observed an increased domain of AP in the maxillary region of *Fgfr1* conditional mutants compared to heterozygous controls (*Figure 6C*). Collectively, these results suggest the FGF repressed, PI3K-dependent differentiation program identified in MEPMs is conserved in vivo.

## Discussion

Our studies show that PDGF and FGF signaling in MEPMs regulate different gene expression programs and phenotypic outputs, with PDGF mainly promoting cell differentiation through PI3K and FGF favoring cell proliferation through ERK. Although the initial wave of gene expression shows high overlap, FGF elicits a quantitatively stronger response in terms of both signal magnitude and duration, which is reinforced by a positive signal from the delayed transcriptional wave in response to FGF but not PDGF. Furthermore, FGF-responsive genes are predominantly ERK dependent, while PDGF targets exhibit greater PI3K dependence genome-wide, relationships mimicked at the level of cellular outcome. Finally, we observed correlation between PDGF-PI3K signaling and osteoblast differentiation at E13.5 as well as increased AP staining in *Fgfr1* mutants, indicating the differentiation circuit repressed by FGF signaling in MEPMs is functional in vivo.

The architecture of the transcriptional response to RTK signaling has been well described as three stereotypic waves: an IEG response, a delayed wave providing feedback control, and a late wave determining cellular outcome (*Amit et al., 2007*; *Avraham and Yarden, 2011*). In MEPMs, the magnitude and duration of the IEG wave is much stronger in response to FGF compared to PDGF, suggesting one level of specificity may be achieved through quantitative differences in IEG induction. Indeed, transient and sustained pERK induction, resulting in distinct magnitudes of *Fos* mRNA expression, drive binary responses in c-Fos abundance and activity at the protein level (*Murphy et al., 2002*; *Nakakuki et al., 2010*), delineating one mechanism through which quantitative differences lead to

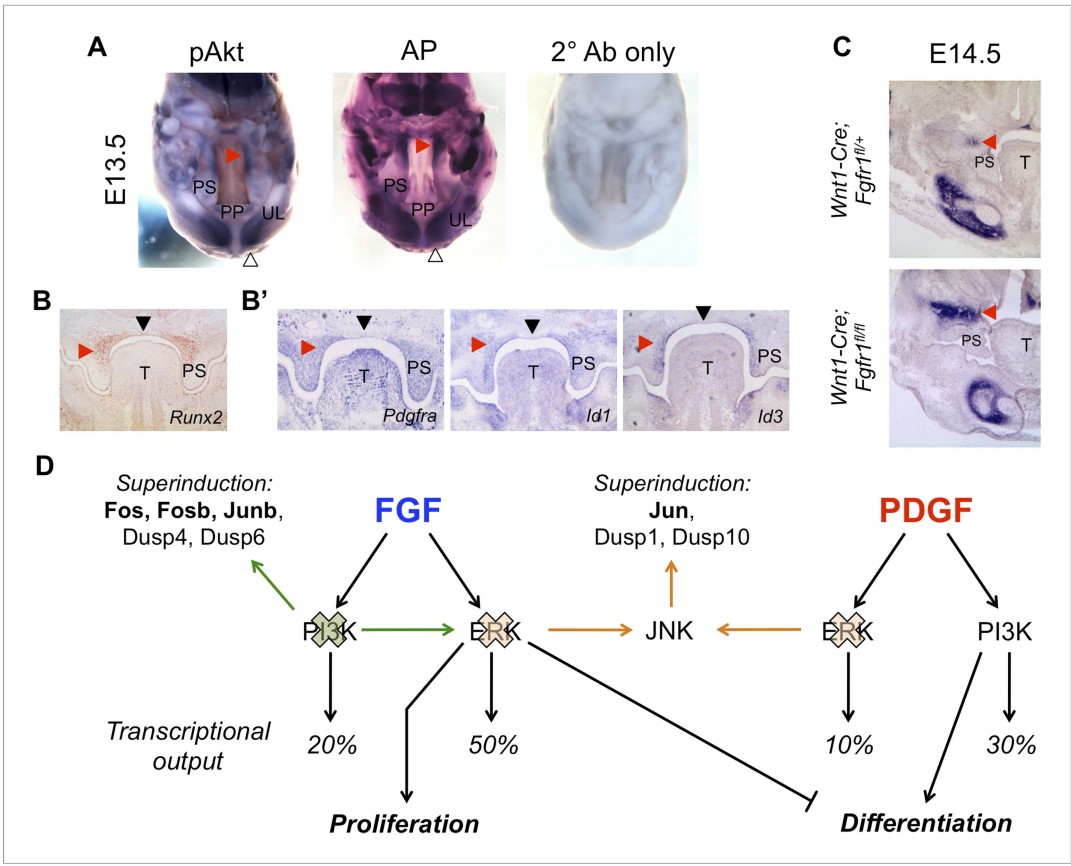

**Figure 6**. In vivo correlation and perturbation of the RTK-mediated differentiation program during mouse craniofacial development. (**A**) At E13.5, pAkt and AP domains co-localize in the middle to posterior secondary palate (red arrowhead) as well as in the developing upper lip (open arrowhead). (**B**) Domain of Runx2 (osteoblast marker) expression overlaps with (**B'**) *Pdgfra*, *Id1*, and *Id3* expression in the middle to posterior palate (red arrowhead), with expression excluded along the midline (black arrowhead). (**C**) Frontal sections from neural crest conditional *Fgfr1* mutants (*Wnt1-Cre; Fgfr1^fl/fl*) exhibit increased AP staining in the developing midface at E14.5, supporting an in vivo role for FGF-mediated repression of osteoblast differentiation (red arrowhead) (n = 3). (**D**) FGF and PDGF signaling use different signaling pathways to instruct divergent cellular outcomes. FGF drives cell proliferation and represses cell differentiation in an ERK-dependent manner, consistent with a greater percentage of the FGF target genes being MEK/ERK dependent (50%) than PI3K dependent (20%). In contrast, PDGF promotes cell differentiation in a PI3K-dependent manner, and PDGF target genes show greater PI3K dependence (30%) than MEK/ERK dependence (10%). Furthermore, inhibition of PI3K signaling leads to an FGF specific induction of pERK (green) and consequently increased transcription of ERK targets such as *Fos*, *Fosb*, and *Junb*. On the other hand, MEK/ERK inhibition leads to pJNK induction (orange) and transcription of *Jun*, indicating multiple crosstalk mechanisms across different intracellular pathways in response to RTK activation. PP: primary palate; PS: palatal shelf; T: tongue; UL: upper lip.

The following figure supplement is available for figure 6:

**Figure supplement 1**. Patterns of gene expression and pERK activity in the E13.5 palate.

a thresholded 'ON-OFF' response in downstream transcriptional activity. Thus, in addition to the observed PDGF-PI3K and FGF-ERK relationships, distinct patterns of IEG transcription factor expression may also contribute to the divergent gene expression profiles at later time points.

While inhibition of the delayed transcriptional response prolongs the PDGF-pERK wave, the opposite effect is observed on FGF-pERK induction. This result is consistent with the diversity of feedback and feedforward regulation on the RTK response (*Avraham and Yarden, 2011*; *Volinsky and Kholodenko, 2013*). In addition to transcriptional feedback, many other mechanisms, including ligand identity and receptor endocytosis (*Francavilla et al., 2013*), contribute toward

specifying the signaling response to RTK activation. The complexity of this regulation underscores the intricate balance between positive and negative control systems, and it will be critical to determine how these regulatory mechanisms interact to produce the desired developmental outcomes.

Given the pleiotropic roles of RTKs, a central question involves how a common set of signal transduction modules specifies distinct outcomes. Our study supports a model in which differential intracellular pathway activation is responsible for the distinct transcriptional responses and cellular processes mediated by RTK signaling in MEPMs (*Figure 6D*). Pathway activation downstream of a single RTK can be affected by many parameters, such as receptor expression level (*Traverse et al., 1994*; *Tallquist et al., 2003*), suggesting that even a single receptor can regulate multiple downstream outputs and transcriptional programs across different contexts. The role of quantitative differences in the pERK response leading to distinct cellular outcomes is well characterized in PC12 neurons (*Marshall, 1995*), and the transcriptional response to the proliferative program mediated by sustained pERK activation in 3T3 fibroblasts has been reported (*Yamamoto et al., 2006*). Building on these studies, the MEPM RNA-seq data provide insight into the differences in gene expression following both transient and sustained pERK induction as well as qualitatively specific target genes downstream of MEK/ERK and PI3K signaling, with the added advantage of profiling these responses within a system requiring these pathways for normal development. It is important to note that mRNA levels alone (as measured by RNA-seq) are not the only level of specificity in the transcriptional response, for example, differential cofactor recruitment can also specify distinct gene expression programs in response to FGF and PDGF signaling (*Vasudevan and Soriano, 2014*). Further work is necessary to identify the precise mechanisms regulating the activity of individual RTK target genes.

Although we focused on MEK/ERK and PI3K signaling based on their reported roles downstream of PDGF and FGF during development (*Klinghoffer et al., 2002*; *Corson et al., 2003*), many other intracellular pathways are activated by RTKs, such as JNK, p38, Src, PLCγ, and PKC (*Lemmon and Schlessinger, 2010*). As previously reported in other contexts (*Bhalla et al., 2002*; *Santos et al., 2007*), we found the FGF-pERK induction was dependent on PKC activity, and we further identified crosstalk in the presence of pathway inhibition between ERK, PI3K, and JNK. In addition, a subset of genes (34% FGF, 43% PDGF) are not affected by MEK/ERK or PI3K inhibition, suggesting they either lie downstream of other intracellular pathways or require inhibition of both ERK and PI3K. Thus, while our studies demonstrate PI3K is necessary for differentiation and ERK is necessary for proliferation in MEPMs, the high connectivity across intracellular pathways suggests no single pathway in isolation is sufficient to drive transcriptional responses and cellular outcomes in their entirety. Rather, integrated output from multiple effector pathways likely contributes to the ultimate cellular outcome, and instead of the existence of linear PDGF-PI3K-differentiation and FGF-ERK-proliferation axes, we favor a model in which multiple signaling events converge on PI3K to promote differentiation and ERK to drive proliferation in the midface.

The finding that effector inhibition increased the induction of many RTK target genes was surprising given their presumed primary function as positive effectors of signaling. The 'superinduction' of these transcriptional responses was due at least in part to compensatory activation of other pathways: inhibition of pERK downstream of both RTKs resulted in increased pJNK induction while inhibition of PI3K downstream of FGF, but not PDGF, resulted in increased pERK activation. This observation has implications for both genetic studies in mice and any system in which effector inhibition is used as an experimental or therapeutic agent. First, allelic series experiments in which adaptor binding is abrogated in order to abolish specific effector cascades may result in induction of alternate signaling pathways, complicating assignment of developmental function to a single intracellular pathway. Indeed, mice harboring a PI3K binding site mutation at the PDGFRα locus show increased SHP2 binding and altered pERK activation (*Klinghoffer et al., 2002*). Second, chemical inhibition of these effectors may prime activation of other pathways, facilitating alternate signaling mechanisms and cellular outcomes. Many examples of such crosstalk between ERK and PI3K (*Mendoza et al., 2011*; *Sun and Bernards, 2014*) as well as ERK and JNK (*Lopez-Bergami et al., 2007*) have been described downstream of RTK signaling in cancer. The MEPM RNA-seq data provide a transcriptional signature for this crosstalk, offering an additional readout to measure activation of these pathways. However, it is important to caution that these responses likely vary based on cellular context, and although the core target genes and effector dependencies may be conserved, extrapolating this framework to other systems requires careful validation.

In comparing our gene expression studies to mouse craniofacial development in vivo, we found co-localization of pAkt and AP in the upper lip and middle to posterior palate at E13.5 as well as

overlapping domains of *Runx2* and *Id* gene expression, indicating the PDGF-PI3K axis identified in MEPMs is correlated with osteoblast differentiation in the midface. It is interesting to note that the distribution of *Pdgfra* mRNA expression compared to pAkt activity and AP staining is not strictly one-to-one; we speculate other factors such as ligand distribution, availability of intracellular signaling proteins, and input from other pathways may contribute toward the spatially restricted domains of osteoblast differentiation. We further observed elevated AP expression in *Fgfr1* conditional mutants at E14.5, consistent with FGF-mediated repression of osteoblast differentiation. Many functions have been ascribed to FGF signaling in skeletogenesis (*Ornitz and Marie, 2002*), and a combination of parameters is likely responsible for the multiplicity of observed roles, such as ligand identity (*Francavilla et al., 2013*) and cellular context. During palatogenesis, *Fgfr1* mutants have been previously reported to exhibit proliferation defects and increased BMP (Bone Morphogenetic Protein) signaling (*Wang et al., 2013*), supporting the notion that FGF drives proliferation and represses differentiation in this system. In addition, the facial clefting phenotypes associated with neural crest conditional loss of RAF, MEK, or ERK (*Newbern et al., 2008*) suggest that the FGF-ERK axis is functionally relevant in vivo.

One important point merits further discussion: Are PI3K and ERK induction interpreted the same independently of the stimulus driving effector activity? There is evidence to suggest this is indeed the case, as IGF (Insulin-like growth factor 1)-mediated PI3K/Akt activation is a key regulator of osteoblast differentiation in mesenchymal stem cells (*Xian et al., 2012*), consistent with the PI3K-mediated differentiation outcome in MEPMs. However, this interpretation is likely restricted in large part by cellular context, as MEK/ERK signaling functions as a positive regulator of differentiation in embryonic stem cells (*Ying et al., 2008*), in contrast to the role of MEK/ERK signaling in MEPMs. In skeletal differentiation, our findings of a MEK/ERK-mediated proliferation program and PI3K-mediated differentiation program are in line with previous work analyzing the role of these effectors (*Mansukhani et al., 2005*; *Raucci et al., 2008*; *Miraoui and Marie, 2010*). In addition, a FGF-PKC-AP-1 signaling axis in calvarial osteoblasts has been reported (*Miraoui et al., 2010*), consistent with the MEPM data linking FGF to a robust AP-1 response and FGF-mediated pERK activation to PKC signaling. Delineating the hierarchy between these signals and pathways and determining the exact combinations sufficient to specify particular outcomes will be key questions for future studies. The present work provides a roadmap of the gene expression profiles underlying these cellular behaviors and a transcriptomic view of how two different RTKs lead to distinct outcomes.

## Materials and methods

### Mouse strains

All animal experiments were approved by the Institutional Animal Care and Use Committee at the Icahn School of Medicine at Mount Sinai. Wild-type C57B/6 mice were used to generate E13.5 MEPMs for RNA-seq. *Pdgfra*^*tm11(EGFP)Sor*^ (*Hamilton et al., 2003*), referred to as *Pdgfra-GFP* in the text, were maintained on a C57BL/6 background, and *FGFR1-CFP* mice (to be described elsewhere), *FGFR1*^*tm5.1Sor*^ mice (*Hoch and Soriano, 2006*), referred to as *Fgfr1*^*fl/fl*^ in the text, and *Tg(Wnt1-Cre)11Rth* mice (*Danielian et al., 1998*), referred to as *Wnt1-Cre* in the text, were all maintained on a 129S4 background.

### Tissue culture

Primary MEPM cells were isolated from E13.5 secondary palatal shelves (day of plug: E0.5) as previously described (*Fantauzzo and Soriano, 2014*; *Vasudevan and Soriano, 2014*). Cells were grown in Dulbecco's modified Eagle's medium (GIBCO; Invitrogen, Carlsbad, CA) with 10% fetal bovine serum (FBS; HyClone Laboratories, Logan, UT), 50 U/mL penicillin (GIBCO), 50 μg/mL streptomycin (GIBCO), and 2 mM L-glutamine (GIBCO). Cells were split twice to passage 2 for all experiments included in this study, and serum starvation was conducted in same media as above but with 0.1% FBS instead of 10% FBS. BrdU assays (*Bush and Soriano, 2010*; *Vasudevan and Soriano, 2014*) and AP staining (*Wu et al., 2008*) were performed as described previously.

### RNA-seq

Passage 2 E13.5 MEPMs were serum starved overnight in 0.1% FBS and then treated with either 30 ng/mL PDGFAA (R&D Systems, Minneapolis, MN) or 50 ng/mL FGF1 (Peprotech, Rocky Hill, NJ) + 1 μg/mL heparin (Sigma-Aldrich, St. Louis, MO) for the desired duration. When pathway inhibitors were used, cells were pretreated with either 1 μM PD325901 or 10 μM LY294002 for 30 min prior to growth factor

addition. MEPMs generated from independent litters were used for each set of replicates, and RNA was collected using RNeasy Mini Kit (Qiagen Inc., Valencia, CA) according to manufacturer's protocol before submission to the Mount Sinai Genomics Core (http://icahn.mssm.edu/departments-and-institutes/genomics/about/resources/genomics-core-facility), where RNA was poly-A selected, libraries generated, and sequenced on Illumina HiSeq 2000. Between 25 and 40 million reads per sample were obtained and mapped to the mouse genome (version mm10) using TopHat (*Kim et al., 2013*). Genes were tested for differential expression by Cuffdiff and considered significant at q <0.1 (*Trapnell et al., 2010*). Data for untreated, 1-hr PDGF, and 1-hr FGF treated MEPMs are publicly available at the Gene Expression Omnibus (GEO), Accession GSE61755 (42). Data for 4 hr PDGF, 4 hr FGF, 1 hr PDGF + PD325901, 1 hr PDGF + LY294002, 4 hr PDGF + PD325901, 4 hr PDGF + LY294002, 1 hr FGF + PD325901, 1 hr FGF + LY294002, 4 hr FGF + PD325901, and 4 hr FGF + LY294002 samples are publicly available at the GEO, Accession GSE66484.

## RT-qPCR

E13.5 MEPMs were serum starved overnight in 0.1% FBS and then treated with desired growth factors and/or inhibitors as for RNA-seq. Total RNA was collected with the RNeasy Mini Kit (Qiagen Inc.). First-strand cDNA was then synthesized using a ratio of 2:1 random primers: Oligo(dT) with SuperScript II RT (Invitrogen). qPCR was performed using a Bio-Rad iQ5 Multicolor Real-Time PCR Detection System and analyzed with iQ5 Optical System Software (version 2.0; Bio-Rad, Hercules, CA). Reactions were performed with PerfeCTa SYBR Green FastMix for iQ (Quanta Biosciences Inc., Gaithersburg, MD) using 10 µM primers (Integrated DNA Technologies Inc., Coralville, IA). A list of qPCR primers used is available in *Supplementary File 5*. The following cycling protocol was used: step 1: 95°C for 3 min; step 2: 95°C for 10 s; step 3: 60° for 40 s; repeat to step 2 39× (total of 40 cycles), and a melting curve analysis was performed. In addition, PCR products were run on a 1.0% agarose gel to ensure correct amplicon size. $\beta 2m$ was used as an endogenous control.

## Western blot

MEPMs were serum starved overnight and treated as for RNA-seq and then washed 3× in ice-cold PBS (Phosphate-buffered saline) before being harvested in NP-40 lysis buffer (20 mM Tris–HCl pH 8, 150 mM NaCl, 10%glycerol, 1% Nonidet P-40, 2 mM EDTA (Ethylenediaminetetraacetic acid), 1× complete Mini protease inhibitor cocktail [Roche Applied Science, Indianapolis, IN], 1 mM PMSF (Phenylmethanesulfonylfluoride), 10 mM NaF, 1 mM Na3VO4, and 25 mM β-glycerophosphate). Total cell lysates were sonicated briefly and then collected by centrifugation. Lysates were then resuspended in Laemmli buffer containing 10% β-mercaptoethanol, heated at 95°C for 5 min, and separated by SDS-polyacrylamide gel electrophoresis.

The following inhibitors were used: LY294002 (Sigma-Aldrich), PD325901 (Stemgent, Cambridge, MA), cycloheximide (Fisher Scientific, Waltham, MA), and Bim I (Santa Cruz Biotechnology, Dallas, TX).

The following antibodies were used: anti-phospho MAPK p42/p44 (9201; Cell Signaling Technologies, Danvers, MA; 1:1000), anti-pAkt (9271; Cell Signaling Technologies; 1:1000), and anti-pJNK (4671; Cell Signaling Technologies; 1:1000). The anti-β tubulin E7 antibody (1:1000) developed by M. Klymkowsky was obtained from the Developmental Studies Hybridoma Bank developed under the auspices of the NICHD and maintained by The University of Iowa, Department of Biology, Iowa City, IA.

## Immunofluorescence

Immunofluorescence was performed as described previously (*Vasudevan and Soriano, 2014*). Briefly, cells were fixed in 4% formaldehyde at room temperature, blocked in 10% donkey serum, stained with primary antibody, and detected with secondary antibody, all for 1 hr at room temperature. The following antibodies were used: Anti-Cleaved Caspase-3 (CST9661; Cell Signaling Technology; 1:400). The anti-BrdU (G3G4, 1:500) developed by S. Kaufman was obtained from the Developmental Studies Hybridoma Bank, created by the NICHD of the NIH and maintained at The University of Iowa, Department of Biology, Iowa City, IA.

## IHC

IHC was performed as described previously (*Fantauzzo and Soriano, 2014*). For whole mounts, embryos were dissected onto ice-cold PBS, fixed overnight in 4:1 methanol:DMSO, cleared in 4:1:1 methanol:DMSO:H$_2$O$_2$, and stored in 100% methanol. For sections, embryos were fixed overnight in

4% paraformaldehyde (PFA) and embedded in paraffin. Staining was done with primary antibody overnight and a goat anti-rabbit IgG peroxidase-conjugated secondary antibody (Jackson ImmunoResearch Laboratories Inc., West Grove, PA). Detection was performed using the Vector Laboratories SK-4100 kit (Vector Laboratories Inc., Burlingame, CA).

## In situ hybridization

Embryos were dissected in ice-cold PBS and fixed overnight in 4% PFA and embedded in paraffin or optimal cutting temperature compound for sectioning. In situ hybridization was performed according to standard protocols (*He and Soriano, 2013*; *Vasudevan and Soriano, 2014*). A list of probe sequences is provided in *Supplementary File 5*.

## Data analysis

Generation of heatmaps, PCA, and other analysis were performed through custom code in R. PCA was done using the 'prcomp' function on centered, median normalized data, and correlation matrix was constructed using the 'cor' function on $\log_2$(FPKM + 1) transformed data. Hierarchical clustering and heatmaps were generated using the 'heatmap.2' and 'hclust' functions. Pearson's correlation was used in the correlation matrix, and Euclidean distance was used as a distance metric for hierarchical clustering. PPI networks were constructed using the Expression2Kinase software (*Chen et al., 2012*) based on an updated version of Genes2Networks (*Berger et al., 2007*), only direct connections (path length = 1) were considered, and all published PPI databases except predicted PPIs were included. Networks were visualized and formatted in yEd (www.yworks.com). For Gene ontology analysis (*Huang et al., 2009*), only those terms at a significance threshold of $p < 0.001$ were included. Redundant GO terms comprising an identical or fully shared subset of genes were removed. A full list of GO results is provided in *Supplementary File 4*.

## Acknowledgements

We are very grateful to Avi Ma'ayan for advice and thoughtful comments on data analysis and visualization. We thank Tony Chen, Aryel Heller, and Anne Levine for excellent technical support; Omar Jabado, Yumi Kasai, and Milind Mahajan for advice and discussions on RNA-seq experiments performed at the Mount Sinai Genomics Core; and members of our laboratory, Andy Chess and Stu Aaronson for helpful discussion and critical comments on this manuscript.

## Additional information

### Funding

| Funder | Grant reference | Author |
|---|---|---|
| National Institutes of Health | R01DE022363 | Philippe Soriano |
| National Institutes of Health | R01DE022778 | Philippe Soriano |
| National Institutes of Health | NRSA Individual Predoctoral Fellowship F31DE023456 | Harish N Vasudevan |
| National Institutes of Health | K99DE024617 | Fenglei He |

The funder had no role in study design, data collection and interpretation, or the decision to submit the work for publication.

### Author contributions

HNV, PM, Conception and design, Acquisition of data, Analysis and interpretation of data, Drafting or revising the article; FH, Acquisition of data, Analysis and interpretation of data, Drafting or revising the article; PS, Conception and design, Analysis and interpretation of data, Drafting or revising the article

### Ethics

Animal experimentation: This study was performed in strict accordance with the recommendations in the Guide for the Care and Use of Laboratory Animals of the National Institutes of Health. All of the animals were handled according to approved Institutional Animal Care and Use Committee (IACUC) protocols (#11-00243) of Icahn School of Medicine at Mt. Sinai.

# Additional files

## Supplementary files

- Supplementary file 1. FPKM values for all genes across all sequenced conditions.
- Supplementary file 2. Expression for all differentially expressed genes (*Cuffdiff* q < 0.1).
- Supplementary file 3. Inhibitor dependence for RTK target genes.
- Supplementary file 4. Compiled gene ontology analysis.
- Supplementary file 5. PCR primers and in situ hybridization probes.

## Major datasets

The following datasets were generated:

| Author(s) | Year | Dataset title | Dataset ID and/or URL | Database, license, and accessibility information |
|---|---|---|---|---|
| Vasudevan HN | 2014 | PDGF and FGF treatment in E13.5 MEPMs | http://www.ncbi.nlm.nih.gov/geo/query/acc.cgi?acc=GSE61755 | Publicly available at NCBI Gene Expression Omnibus (GSE61755). |
| Vasudevan HN | 2015 | PDGF and FGF treatment in E13.5 MEPMs II | http://www.ncbi.nlm.nih.gov/geo/query/acc.cgi?acc=GSE66484 | Publicly available at NCBI Gene Expression Omnibus (GSE66484). |

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
