## [Decision Letter]

Thank you for sending your work entitled “Receptor tyrosine kinases modulate distinct transcriptional programs by differential usage of intracellular pathways” for consideration at *eLife*. Your article has been favorably evaluated by Fiona Watt (Senior editor), Marianne Bronner (Reviewing editor), and three reviewers.

The Reviewing editor and the reviewers discussed their comments before we reached this decision, and the Reviewing editor has assembled the following comments to help you prepare a revised submission.

In this study, the authors took advantage of palatal mesenchymal cells in which both FGF and PDGF signaling utilize receptor tyrosine kinase (RTK) to regulate downstream target genes during palatogenesis. To investigate the mechanisms underlying RTK specificity in craniofacial development, the authors performed detailed FGF and PDGF responsive gene analyses. They show that FGF signaling favors cell proliferation through activation of ERK whereas PDGF signaling mainly promotes cell differentiation through PI3K. Their results show that PDGF requires PI3K and affects differentiation while FGF depends on MEK/ERK and promotes proliferation. Furthermore although the FGF response was shown to be quantitatively stronger, there was considerable interconnectivity across the intracellular pathways which urges caution with respect to general effector inhibitor experiments and conclusions. The study is carefully done. Some further experimentation, described below, would strengthen the results.

1) The interpretations seem too focussed on the differences in MEK and PI3K signaling as explaining the changes in transcription between 1h and 4h. For example, in the Results section it is concluded: “In sum, both RTKs induce a qualitatively similar gene expression program at 1 hour, but FGF drives a quantitatively stronger early response, resulting in divergent gene expression profiles at 4 hours.” This seems to be jumping the gun, as much of the evidence comes later in the Results. What is meant by ‘similar’ gene expression programs at early stages? Perhaps there are some differences in early transcription factor expression that contribute to the greater divergence at 4h. A network of induced genes is shown in Figure 1, but are these the only ones? It would be useful to have a list of the differentially expressed genes at 1h, and some explicit discussion on this point.

2) Although the PDGF-PI3K/Akt signaling axis and FGF-MEK/ERK signaling are known to be important for craniofacial development, this concern is outweighed by the novelty of the comparative systematic transcriptional analyses of PDGF and FGF signaling in the palatal mesenchyme under the same experimental conditions. Some deeper analyses of the respective *Fgfr1* or *Pdgfra* mutants (either null or conditional as appropriate) would strengthen the authors' interpretations and conclusions in vivo. The RTK pathway intersection the authors observed in MEPM cells could easily be thoroughly validated in the background of the respective mutants.

3) In the experiment where the authors inhibited pERK by using PD325901, the authors found activation of pJNK. It would be helpful to know if there are any changes in apoptotic activity in PD325901-treated MEPMs.

4) In reviewing the *Pdgfra* expression pattern, it is clear that *Pdgfra* is localized throughout the palatal mesenchyme (Figure 6) whereas the condensation of palatal mesenchyme and osteogenic differentiation is mainly located in the nasal region of the palatal shelf (Figure 6). What is the functional significance of PDGF signaling in the oral region of the palatal shelf? Is PI3K activation only restricted to the nasal region where osteogenic differentiation will occur?

5) The tissue orientation in Figure 6 is not clear. In general, palatal shelves should have fused at E14.5 in the control sample. Here, both control and *Fgfr1* mutant samples show that palatal shelves are blocked by the tongue.

---

## [Author Response]

*1) The interpretations seem too focussed on the differences in MEK and PI3K signaling as explaining the changes in transcription between 1h and 4h. For example, in the Results section it is concluded: “In sum, both RTKs induce a qualitatively similar gene expression program at 1 hour, but FGF drives a quantitatively stronger early response, resulting in divergent gene expression profiles at 4 hours.” This seems to be jumping the gun, as much of the evidence comes later in the Results. What is meant by ‘similar’ gene expression programs at early stages? Perhaps there are some differences in early transcription factor expression that contribute to the greater divergence at 4h. A network of induced genes is shown in*
Figure 1*, but are these the only ones? It would be useful to have a list of the differentially expressed genes at 1h, and some explicit discussion on this point*.

We agree that a more thorough discussion of the transcription factor changes observed at 1 hour is warranted. By ‘similar’ gene expression programs, we are referring to the high correlation coefficient between the fold change of differentially regulated genes at 1 hour for both PDGF and FGF treatment, as plotted in Figure 1, compared to the correlation at 4 hours, as plotted in Figure 1. Further, the set of PDGF targets at 1 hour are a subset of FGF targets at 1 hour (i.e. all PDGF targets at 1 hour are also identified as significantly regulated FGF targets by Cuffdiff, although the kinetics and magnitude of regulation are different). The main text has been amended in the Results section to make these points explicit. In addition, of the 34 total transcription factors identified within the FGF target gene list at 1 hour, 76% (26/34) are also significantly regulated by PDGF stimulation ([Supplementary-material SD4-data]), again underscoring the high degree of overlap at the early timepoint.

We do concur with the reviewers' point that quantitative differences in the early transcriptional response (which is shared across both PDGF and FGF stimulation) likely contribute to the divergence at 4 hours. Indeed, there exist examples in which the magnitude and duration of IEG induction can specify distinct late responses (Murphy, LO, et al. Nat. Cell Biology. 4, 556-564, 2002 and Nakakuki, T. et al. Cell. 141, 884–896, 2010). We accordingly altered our statement in the Results section, as the reviewers rightly indicate much of the data comes later in the manuscript, and we further added a few sentences in the Discussion summarizing the likely importance of distinct transcription factor induction patterns (in addition to MEK/ERK and PI3K dependence) to explain the observed differences at 4 hours.

In the presented PPI network (Figure 1), 25 of the 113 genes significantly upregulated at 1 hour are depicted, and this statement has been added to the main text (Results section). A full list of these genes can be found in [Supplementary-material SD2-data], Columns A-H on the sheet entitled ‘ComparedToUntreated.’

*2) Although the PDGF-PI3K/Akt signaling axis and FGF-MEK/ERK signaling are known to be important for craniofacial development, this concern is outweighed by the novelty of the comparative systematic transcriptional analyses of PDGF and FGF signaling in the palatal mesenchyme under the same experimental conditions. Some deeper analyses of the respective Fgfr1 or Pdgfra mutants (either null or conditional as appropriate) would strengthen the authors' interpretations and conclusions in vivo. The RTK pathway intersection the authors observed in MEPM cells could easily be thoroughly validated in the background of the respective mutants*.

We agree that in vivo validation of our observations in MEPMs is highly desirable. Neither Pdgfra (Soriano, P. Development. 124, 2691-2700, 1997) nor Fgfr1 (Deng, CX. et al. Genes and Dev. 8, 3045-3057, 1994. and Yamaguchi, TP. et al. Genes and Dev. 8, 3032-3044, 1994) null mutants are recovered at Mendelian ratios at E13.5. Thus, we carried out RT-qPCR analysis in E13.5 secondary palates isolated from Wnt1-Cre; Pdgfra^fl/fl^ and Wnt1-Cre; Fgfr1^fl/fl^ neural crest conditional mutants (Figure 7).

Author response image 1.Expression of PDGF and FGF target genes in Pdgfra (red) and Fgfr1 (blue) mutant E13.5 secondary palates compared to control embryos without the Wnt1-Cre transgene (black). *p < 0.1; **p < 0.01, n = 4 for each genotype.**DOI:**
http://dx.doi.org/10.7554/eLife.07186.021

The expression of IEGs such as Fos and Jun is unchanged, an expected result given the number of stimuli capable of inducing these genes as well as the observation that both PDGF and FGF treatment can induce these genes in MEPMs. Many differentiation genes (Mef2c, Alx1, Alx4, and Alpl) exhibit significantly greater expression in Fgfr1 mutants compared to Pdgfra mutants or control embryos, consistent with the MEPM data in which these genes display greater expression at 4 hours following PDGF treatment compared to 4 hours following FGF treatment. However, the Id genes (Id1, Id2, and Id3) do not conform to this trend and have similar expression across all genotypes; this may be due to compensation in vivo, particularly by BMP signaling, a canonical inducer of Id gene expression. Finally, it is interesting to note a high level of residual Pdgfra and Fgfr1 expression in the corresponding mutants. We believe this reflects receptor expression from non-neural crest derived contributions to the secondary palate (such as the mesoderm). Indeed, when we performed in situ hybridization for Pdgfra in Wnt1-Cre; Pdgfra^fl/fl^ mutants and Fgfr1 expression in Wnt1-Cre; Fgfr1^fl/fl^ mutants, we found some receptor mRNA expression in the E13.5 palate of the respective conditional mutants (data not shown), consistent with our qPCR results. Furthermore, flow cytometry analysis of PDGFRα expression in 10.5 branchial arches of Wnt1-Cre; Pdgfra^fl/fl^ conditional mutants revealed no receptor expression, indicating satisfactory deletion by the Wnt1-Cre transgene in early structures derived from the neural crest (Tallquist, MD, and Soriano, P. Development. 130, 507-518, 2003). Finally, the same qPCR primers, when used to measure Pdgfra and Fgfr1 expression in E11.5 facial prominences isolated from the respective conditional mutants, revealed almost complete loss of these genes, thus suggesting the observed residual expression is likely not a technical issue associated with the qPCR protocol itself (Vasudevan, HN, and Soriano, P. Dev. Cell. 31, 332-344, 2014).

In summary, although we do observe supportive trends for a number of differentiation genes (Mef2c, Alx1, Alx4, and Alpl), results are mixed for other targets, which may be explained by compensation in vivo, residual receptor expression, or other factors. The development of Cre drivers more precise than the Wnt1-Cre that drive recombination within the secondary palate itself (rather than in the early neural crest) would facilitate the in vivo validation of the observations made in MEPMs. We believe the generation of such tools and subsequent analysis of Pdgfra and Fgfr1 mutants, while certainly important, is outside the scope of this study. We chose not to include the in vivo data obtained from the Wnt1-Cre mutant palatal shelves in the main text, as the observed residual receptor expression precludes clear interpretation of these results.

*3) In the experiment where the authors inhibited pERK by using PD325901, the authors found activation of pJNK. It would be helpful to know if there are any changes in apoptotic activity in PD325901-treated MEPMs*.

We have performed the suggested experiment and measured apoptosis using cleaved caspase-3 (CC3) for FGF and PDGF treated MEPMs in the presence and absence of both PD325901 (MEK inhibitor) and LY294002 (PI3K inhibitor) (Figure 5—figure supplement 1). In the case of FGF treatment, both MEK and PI3K inhibition increase the percentage of CC3+ cells, but MEK inhibition has a significantly greater effect, consistent with the induction of pJNK in this condition (Figure 4). In contrast, PI3K inhibition has a greater effect on apoptosis in PDGF treated cells, and again, both inhibitors show significantly increased apoptosis compared to PDGF treated cells alone. This result is not wholly surprising given the modest pJNK induction in the LY294002 dose response curve for PDGF treated cells (Figure 4). In addition, both PD325901 and LY294002 are known to induce apoptosis in other cell types, and our results suggest intracellular signaling in response to FGF and PDGF stimulation exhibit different sensitivities to the apoptotic effect of these inhibitors. The CC3 measurements are generally consistent with the crystal violet assay, where FGF exhibits greater MEK dependence while PDGF-mediated effects on cell viability are sensitive to both MEK and PI3K inhibition (Figure 5—figure supplement 1). We have added this result in Figure 5—figure supplement 1 and removed the schematic of RTK mediated control of cell proliferation and differentiation, as a similar model is presented in Figure 6. We have included a line in the main text describing these apoptosis results (please see the subsection headed “FGF drives MEK/ERK-dependent cell proliferation, and PDGF promotes PI3K-dependent cell differentiation”).

*4) In reviewing the Pdgfra expression pattern, it is clear that Pdgfra is localized throughout the palatal mesenchyme (*Figure 6*) whereas the condensation of palatal mesenchyme and osteogenic differentiation is mainly located in the nasal region of the palatal shelf (*Figure 6*). What is the functional significance of PDGF signaling in the oral region of the palatal shelf? Is PI3K activation only restricted to the nasal region where osteogenic differentiation will occur?*

The reviewers astutely observe that Pdgfra mRNA expression is evenly distributed, which leads to a question regarding the spatial distribution of Pdgfra mRNA expression, PI3K activation, and osteogenic differentiation in the palate. Regarding PI3K activation, it is important to note that our immunohistochemistry results identify regions of relative enrichment for pAkt and may not reflect an absolute lack of pAkt in regions that appear negative. In addition, PI3K can signal through Akt independent pathways, and it is possible PI3K activation recruits these other pathways outside the observed pAkt domains. While Pdgfra mRNA expression is ubiquitous throughout the palatal mesenchyme, receptor expression level, ligand distribution, or availability of intracellular signaling proteins may all affect the extent and function of PDGF signaling in different regions. It is also possible that PDGF signaling is necessary but not sufficient for PI3K activation and osteogenic differentiation. In this scenario, the spatially restricted patterns of pAkt induction and osteogenic differentiation may function as coincidence detectors that respond only to the coordinate activation of multiple signals. We have added a line in the Discussion mentioning these possibilities. One approach to unambiguously determine the functional role of PDGFRα in the oral region could involve the use of a spatially specific Cre driver, but the tools to perform such an analysis are currently lacking.

*5) The tissue orientation in*
Figure 6
*is not clear. In general, palatal shelves should have fused at E14.5 in the control sample. Here, both control and Fgfr1 mutant samples show that palatal shelves are blocked by the tongue*.

The reviewers raise a concern regarding the time of palatal fusion in mouse development. In our mice, which are maintained on a 129S4 background, secondary palate fusion is not complete at E14.5 (see also Fantauzzo, KA, and Soriano, P. Genes Dev. 28, 1005–1017, 2014). This is also consistent with previous observations made with SEM (Kaufman. The Atlas of Mouse Development. 426-429, 1992) and H/E staining (Bush, JO, and Jiang, R. Development 139, 828–828, 2012). However, it is intriguing to note that other databases, such as FaceBase depict contact between the palatal shelves at E14.5 (https://www.facebase.org/sites/default/files/SEM_images/anatomy.html). We suspect this discrepancy in the exact time of secondary palate fusion may be background dependent, as FaceBase uses C57BL/6 embryos. Regarding the sections from Fgfr1 mutant palatal shelves, we have replaced the original image with a lower magnification image from a different embryo (Figure 6) to better orient the reader and have clarified in the figure legend that these are frontal sections.